# Mesenchymal Stem Cell Use in Acute Tendon Injury: In Vitro Tenogenic Potential vs. In Vivo Dose Response

**DOI:** 10.3390/bioengineering9080407

**Published:** 2022-08-22

**Authors:** Kristin Bowers, Lisa Amelse, Austin Bow, Steven Newby, Amber MacDonald, Xiaocun Sun, David Anderson, Madhu Dhar

**Affiliations:** 1Large Animal Clinical Sciences, University of Tennessee College of Veterinary Medicine, Knoxville, TN 37996-4550, USA; 2Office of Information and Technology, University of Tennessee, Knoxville, TN 37996, USA

**Keywords:** mesenchymal stem cell, tendon, transforming growth factor beta-3, connective tissue growth factor, extracellular matrix, tenogenic differentiation

## Abstract

Stem cell therapy for the treatment of tendon injury is an emerging clinical practice in the fields of human and veterinary sports medicine; however, the therapeutic benefit of intralesional transplantation of mesenchymal stem cells in tendonitis cases is not well designed. Questions persist regarding the overall tenogenic potential and efficacy of this treatment alone. In this study, we aimed to isolate a rat mesenchymal stem cell lineage for in vitro and in vivo use, to assess the effects of growth factor exposure in vitro on cell morphology, behavior, and tendon-associated glycoprotein production, and to assess the therapeutic potential of intralesional stem cells, as a function of dose, in vivo. First, rat adipose-derived (rAdMSC) and bone marrow-derived (rBMSC) stem cell lineages were isolated, characterized with flow cytometric analysis, and compared in terms of proliferation (MTS assay) and cellular viability (calcein AM staining). Rat AdMSCs displayed superior proliferation and more homogenous CD 73, CD 44H, and CD 90 expression as compared to rBMSC. Next, the tenogenic differentiation potential of the rAdMSC lineage was tested in vitro through isolated and combined stimulation with reported tenogenic growth factors, transforming growth factor (TGF)-β3 and connective tissue growth factor (CTGF). We found that the most effective tenogenic factor in terms of cellular morphologic change, cell alignment/orientation, sustained cellular viability, and tendon-associated glycoprotein upregulation was TGFβ3, and we confirmed that rAdMSC could be induced toward a tenogenic lineage in vitro. Finally, the therapeutic potential of rAdMSCs as a function of dose was assessed using a rat acute Achilles tendon injury model. Amounts of 5 × 10^5^ (low dose) and 4 × 10^6^ (high dose) were used. Subjectively, on the gross morphology, the rAdMSC-treated tendons exhibited fewer adhesions and less scar tissue than the control tendons; however, regardless of the rAdMSC dose, no significant differences in histological grade or tissue collagen I deposition were noted between the rAdMSC-treated and control tendons. Collectively, rAdMSCs exhibited appropriate stem cell markers and tenogenic potential in vitro, but the clinical efficacy of intralesional implantation of undifferentiated cells in acute tendonitis cases could not be proven. Further investigation into complementary therapeutics or specialized culture conditions prior to implantation are warranted.

## 1. Introduction

Stem cell therapy for the treatment of tendon injury is an emerging clinical practice in the fields of human and veterinary sports medicine. Intralesional stem cell implantation, for example, has been utilized for tendonitis treatment in sport horses for over a decade with promising results [1]. In a report of 105 National Hunt horses with over two years of follow-up, there was less recurrence of tendonitis in those horses treated with stem cells as opposed to traditional therapies [2]. An analysis of several case-control observational studies in equine sport horses indicated that the treatment of tendonitis with bone marrow derived mesenchymal stem cells (BMSC) was associated with a reduction in the rate of reinjury [3,4,5]. Further clinical research is necessary to elucidate the potential of stem cell therapy in cases of acute or chronic tendon injury. In an experimental model of Achilles injury using rats, tissue treated with BMSCs exhibited superior biomechanical strength and histological evidence of healing when compared to control samples [6]; however, the therapeutic benefit of intralesional transplantation of mesenchymal stem cells in tendonitis cases is disputed in the literature, with reports of no significant difference between BMSC-treated and control tendons in other Achilles injury trials [7,8,9]. Several strategies have been proposed to elevate the therapeutic potential of intralesional stem cell therapy including predifferentiation, biomaterial-based cell delivery, or concurrent growth factor injection [10,11,12,13], but uncertainty regarding the ideal therapeutic strategy in terms of cell source, expansion characteristics, and cell numbers persists [13,14,15].

Unlike osteogenic, chondrogenic, and adipogenic differentiation, a standardized, effective protocol for the tenogenic differentiation of mesenchymal stem cells has not been established [16,17,18]. Methods include growth factor exposure, dynamic mechanical stimulation, biomaterial/ECM scaffold three-dimensional culture, and coculture with tendon-derived stem cells or tenocytes [17,19,20]. Growth factors that have been positively correlated with tendon associated biomarkers include transforming growth factor beta ligands (TGFβ), specifically TGFβ1 and TGFβ3, bone morphogenic protein (BMP) ligands, specifically BMP-12, BMP-13, and BMP-14, fibroblast growth factor (FGF) ligands, specifically FGF-2, and connective tissue growth factor (CTGF). These cytokines are inherently linked, and they exhibit significant co-expression through the mitogen-activated protein kinase (MAPK) pathway during embryonic development and active homeostasis/tissue turnover in adults [21,22]. Recently, Yin et al. investigated the tenogenic effects of TGFβ-1, BMP-12, and CTGF/ascorbic acid exposure on mesenchymal stem cells and noted superior morphologic changes, genetic upregulations, and tendon-associated marker production using a stepwise protocol of initial TGFβ-1 exposure, followed by CTGF/ascorbic acid [23]. In addition, several studies have reported similarly promising results following MSC exposure to TGFβ-3 alone in either a monolayer or three-dimensional culture [20,24,25,26,27].

While uncertainty as to the ideal cell type and predifferentiation protocol for intralesional MSC therapy in tendon injury persists, another potential contributor to its inconsistent therapeutic benefit is the lack of an established therapeutic dose of mesenchymal stem cells [28,29,30,31]. Initially, MSCs were thought to aid healing directly through differentiation into tissue-appropriate cells, replication, and the reestablishment of tissue architecture, but more recent research has suggested that their function is more immunomodulatory than direct repopulation. Increasingly, experimental studies and clinical practice favor the use of allogenic MSCs, rather than autologous MSCs [28,32]. Allogenic MSCs provide the benefits of proven cellular purity, multipotency, consistency, and ease of acquisition without the delays required for autologous cell isolation, characterization, and propagation [33,34,35]. These benefits typically outweigh the risk of an inflammatory (rejection) reaction in response to foreign proteins, specifically major histocompatibility complexes (MHC) [35,36]; however, too high of a dose of allogenic MSCs may impair their immunomodulatory functions by initiating an overwhelming MHC-mismatch inflammatory response [35,37].

Dose-response studies to ascertain an effective dose or an adverse threshold for mesenchymal stem cell therapy are lacking. In one study, Saether et al. compared the effects of high dose (4 × 10^6^ MSCs) and low dose (1 × 10^6^ MSCs) stem cells used in a study of rat medial collateral ligament healing [28]. Improved strength and stiffness of the ligament and a decreased inflammatory response in the low dose group was noted, but neither the high nor the low dose MSC treatments downregulated inflammatory cytokines or restricted inflammatory cell migration when compared to the untreated controls [28]. This study was repeated using a rat Achilles tendon model and introduced the variables of preconditioning and genetic modification to promote the production of anti-inflammatory modulators [35]. The authors reported that anti-inflammatory preconditioning through genetic engineering had a more significant effect on tendon healing than dosage in terms of the M1 macrophage number and amount of inflammatory IL-12 cytokine. Despite these reports demonstrating in vivo efficacy of MSCs for tendon healing and repair, these therapies cannot be directly translated into the clinic. This is because of the regulatory constraints of using genetically-modified MSCs. Furthermore, the use of minimally-modified MSCs does show promise and, hence, for translation research, undifferentiated, naïve adult tissue-derived MSCs are preferred. Additionally, the effect of the MSC dose alone on tendon injury healing has not been described and an effective dose of MSC for use in the treatment of acute tendonitis has not been established.

Our goals were to isolate a rat mesenchymal stem cell lineage for in vitro and in vivo use, to assess the effects of growth factor exposure in vitro on cell morphology, behavior, and tendon-associated glycoprotein production, and to assess the therapeutic potential of intralesional stem cells as a function of the dose in vivo. We aimed to characterize and establish the cells’ tenogenic potential in vitro prior to implantation in vivo in a dosing trial. TGFβ-3 was selected for this experiment based on promising published reports when used in vitro for tenogenic differentiation [20,25,26,27]; CTGF with ascorbic acid was selected due to its reported synergy with TGFβ-1 in a stepwise tenogenic differentiation protocol [23]. A rodent model of tendon injury is useful for an assessment of the effect of mesenchymal stem cell dose on in vivo healing with the achievement of appropriate endpoints in a time efficient manner [38,39,40]. The calcaneal tendon injury model is well characterized and provides guidelines specific to the duration of study, creation of defect, and analytical methods [6,7,8]. We hypothesized that cell culture-expanded rat MSCs have a potential to undergo tenogenesis in vitro, and this can be evaluated by concurrent exposure to TGFβ-3, CTGF, and ascorbic acid. We also hypothesized that intralesional MSC implantation would have a dose-dependent and beneficial effect on tendon healing in an acute injury model.

## 2. Materials and Methods

### 2.1. Ethics

All studies were reviewed and approved by the University of Tennessee Institutional Animal Care and Use Committee and adhered to the National Institutes of Health’s Guide for the Care and Use of Laboratory Animals [41].

### 2.2. Isolation, Culture, and Characterization of Rat Adipose-Derived MSCs

All MSCs were isolated from Sprague-Dawley rats (150–200 g). BMSCs were collected from the bone marrow of the femurs as previously described [42,43,44,45]. Briefly, the femurs were rinsed with phosphate-buffered saline (PBS) before clipping the ends of the bone. An 18-gauge needle was used to extrude the bone marrow, and marrow was transferred to a collection tube using growth media flushes. After centrifugation at 1000 rpm for 5 min, the cell pellet was re-suspended in 5.5 mL growth media and filtered with a 70 μm strainer twice to ensure complete collection. The AdMSCs were collected from perinephric and testicular adipose tissue as previously described [42,43]. The tissue was minced and digested in PBS containing a 0.1% Type I Collagenase Buffer (Sigma-Aldrich, St. Louis, Mo, USA) at 37 °C with intermittent mixing; incubation was discontinued when the tissue appeared homogenous (roughly 60 min), and growth media was added to inactivate the collagenase. After centrifugation at 600 rpm for 5 min, the lipid layer and collagenase/media liquid fraction were removed, and the cell pellet was resuspended in PBS. Centrifugation and resuspension were repeated, and the cells were filtered through a 70 μm strainer. Isolated bone marrow and adipose cells were cultured in a complete medium consisting of Dulbecco’s Modified Eagle Medium (DMEM-F12, Thermo Fisher Scientific, Waltham, MA, USA) containing 10% fetal bovine serum (FBS, Thermo Fisher Scientific, Waltham, MA, USA), 1% penicillin–streptomycin–neomycin (Thermo Fisher Scientific, Waltham, MA, USA), and 0.1% amphotericin B (Thermo Fisher Scientific, Waltham, MA, USA). The cells were grown to 80–90% confluency and then harvested with 0.05% trypsin for cryopreservation (50% FBS, 45% DMEM-F12, and 5% DMSO) or re-seeded for expansion in the complete medium. 

### 2.3. Immunophenotyping

Flow cytometry analysis was used as previously reported to characterize the isolated rAdMSC and rBMSC [45]. Cells were identified by the expression of the following cell-surface cluster-of-differentiation (CD) markers: CD29, CD73, CD44H, CD90, CD106, CD11b/c, CD45, and CD31. All the markers tested are recognized by the Mesenchymal and Tissue Stem Cell Committee of the International Society for Cellular Therapy [42]. Anti-CD antibodies were used at the manufacturer’s recommended concentrations (Biolegend, San Diego, CA). The labeled cells were assessed using a BD FACS Calibur Flow Cytometer (BD Biosciences, Haryana, India) and FlowJo software (FlowJo LLC, Ashland, OR, USA).

### 2.4. Proliferation Assay

Cell proliferation rates of rat adipose-derived and bone-marrow-derived mesenchymal stem cells were assessed at 1, 3, 5, and 7 days of culture using the CellTiter 96 Aqueous Non-Radioactive (MTS) assay (Promega, Madison, WI, USA) as previously described [46,47]. Briefly, all the experiments were performed using 24-well plates with 1.0 × 10^4^ rBMSC or rAdMSC (passage 5 and passage 6, respectively) seeded in 500 µL of complete growth medium as the seeding volume. The optical density of the cell and MTS reagent complex was measured using a microplate fluorescence reader (BioTek, Winooski, VT, USA) at 490 nm. Medium without cells was used as a blank. Each timepoint was performed in triplicate, and an average absorbance at 490 nm (corrected by the reference blank reading) of the three wells was utilized as data for further analysis. The absorbance data was analyzed for the time effect and coefficients of determination (R^2^ values) of the linear regression were calculated for a direct comparison. At each timepoint, an additional well was utilized for qualitative cell viability analysis using Calcein AM Viability Dye (Thermo Fisher, Waltham, MA, USA); stained cells were visualized after 5 min, and images were obtained using a fluorescent microscope (Leica Microsystems, Wetzlar, Germany).

### 2.5. Tenogenic Differentiation of Rat Adipose-Derived MSCs

Based on the results of the proliferation assay and immunophenotyping, rAdMSCs were utilized for the remainder of the experimentation. Rat AdMSCs (1.0 × 10^6^ cells, passage 3) were removed from liquid nitrogen, thawed, suspended in complete growth medium, seeded into T175 flasks for expansion, and incubated at 37 °C in a humidified atmosphere with 5% CO_2_. The medium was changed every two to three days until the cells reached 70–80% confluence. Upon reaching confluence, adherent cells were harvested using 0.25% trypsin EDTA (Thermo Fisher Scientific, Waltham, MA, USA) and transferred to 6-well plates for the tenogenic differentiation trial at a seeding density of 5.0 × 10^4^ cells in 3 mL of complete media per well. The rAdMSCs were maintained in complete medium alone (Control) or supplemented with (a) 10 ng/mL of TGF-β3 (Peprotech, Rocky Hill, NJ, USA), (b) 100 ng/mL of CTGF (Peprotech, Rocky Hill, NJ, USA) with 50 µg/mL of ascorbic acid (Sigma-Aldrich, St. Louis, MO, USA), or (c) 10 ng/mL of TGF-β3, 100 ng/mL of CTGF, and 50 µg/mL of ascorbic acid. The growth factor concentrations were selected based on previously published reports regarding growth-factor-mediated tenogenic differentiation [15,23,25,26,48]. The conditioned media was changed every two days and the cells were imaged daily using bright light phase-contrast microscopy (Leica Microsystems, Wetzlar, Germany). The cells were monitored for signs of differentiation including changes in cellular morphology, cell alignment/orientation, and cell-to-cell adherence/clumping, and the cell culture was halted if a marked cellular exfoliation (greater than 30% of the previously adhered cells) was noted.

### 2.6. Immunofluorescence

For the immunofluorescence assays, the previously described tenogenic differentiation experiment was repeated using the same growth factors and concentrations of each. Briefly, rAdMSCs (passage 4) were harvested with 0.25% trypsin EDTA (Thermo Fisher Scientific, Waltham, MA, USA) and transferred to 12-well plates for the tenogenic differentiation trial at a seeding density of 1.5 × 10^4^ cells in 1 mL of complete media per well. The rAdMSCs were maintained in complete medium alone (Control) or supplemented with (a) 10 ng/mL of TGF-β3 (Peprotech, Rocky Hill, NJ, USA), (b) 100 ng/mL of CTGF (Peprotech, Rocky Hill, NJ, USA) with 50 µg/mL of ascorbic acid (Sigma-Aldrich, St. Louis, MO, USA), or (c) 10 ng/mL of TGF-β3, 100 ng/mL of CTGF, and 50 µg/mL of ascorbic acid. The conditioned media was changed every two days and the cell morphology was monitored daily using bright light phase-contrast microscopy. On days 1, 4, and 8, cells from each treatment group were fixed with 4% paraformaldehyde at room temperature for 10 min, permeabilized with 0.1% Triton X-100 in a Hank’s Balanced Salt Solution (HBSS), and blocked with Universal Blocking Reagent (BioGenex, Fremont, CA, USA) for 30 min at room temperature. The cytoskeletal organization and cellular morphology were assessed by evaluating the expression of extracellular matrix proteins, F-actin and Collagen I, as previously described [45,49]. The expression profiles of tendon-associated proteoglycans, tenascin C (TenC) and tenomodulin (Tnmd), were also evaluated at each timepoint as markers of tenogenic differentiation [17,20,24,50]. Fixed cells were incubated with 1–2 µg of primary antibodies for F-actin (Alexa Fluor 594 phalloidin, Invitrogen, Waltham, MA, USA), Collagen I (Abcam, Cambridge, UK), TenC (Invitrogen, Waltham, MA, USA), and Tnmd (Biorbyt, Cambridge, UK) at 4 °C for 24 h. Subsequently, the cells were washed with HBSS and incubated with appropriate Alexa Fluor secondary anti-mouse or anti-rabbit antibodies at room temperature for 30 min in dark; Alexa Fluor 594 phalloidin was preconjugated and did not undergo secondary antibody exposure. The cells were mounted with a Prolong Gold antifade reagent with 4′, 6-diamidino-2-phenylindole (DAPI) nuclear stain (Life Technologies, Carlsbad, CA, USA). The cells were analyzed and imaged using a fluorescent microscope (Leica Microsystems, Wetzlar, Germany). The images were further analyzed using Fiji image processing open-source software (ImageJ, Bethesda, MD, USA) to obtain the DAPI-labeled cell counts, average nuclear size, TenC positive cell counts, and Tnmd positive cell counts [51,52].

### 2.7. Fibrin Gel

In order to achieve the local administration of stem cells in a controlled fashion, cells were suspended in fibrin gel, a viscous transport medium that will entrap but not inhibit cells [53]. The fibrin gel was fabricated in our lab using fibrinogen and thrombin (MilliporeSigma, Burlington, MA) at a ratio of 0.2 U thrombin: 1 mg fibrinogen. This technique has been established and validated in our laboratory for cell uptake and suspension within the gel.

### 2.8. Surgery

Twenty 12-week-old female Sprague-Dawley rats each weighing between 208 and 242 g were acquired and maintained in a climate-controlled animal housing. Each rat underwent a left hindlimb Achilles injury and was assigned intralesional treatment with the fibrin gel alone (Control), 5 × 10^5^ rAdMSC in fibrin gel (Low Dose), or 4 × 10^6^ rAdMSC in fibrin gel (High Dose); each treatment group consisted of a minimum of six rats. In order to achieve clinical relevance, undifferentiated, naïve MSCs were used in vivo. The dosages were selected based on those previously employed in rat tendon or ligament injury models [6,7,28,54,55,56], and they reflect the lowest and highest reported dosages utilized in these models. The left hindlimb Achilles injury was induced as previously described [7,8,57]. Briefly, each rat was anesthetized using isoflurane gas and administered preoperative analgesics (buprenorphine 0.02 mg/kg SQ). A 1 cm longitudinal incision was made over the caudolateral aspect of the left hindlimb at the level of the Achilles tendon, terminating immediately proximal to the calcaneal insertion. Then, the central third of the Achilles tendon was isolated using blunt dissection and removed, resulting in a central defect that extended from the myotendinous junction to immediately proximal to the calcaneal insertion. Following the defect creation, the appropriate treatment was administered intralesional using forceps to transfer the semi-solid fibrin gel into the lesion site. The skin incision was closed in a single layer using a full thickness, interrupted cruciate pattern.

### 2.9. Postoperative Management

The animals were individually housed with free cage activity during the postoperative period. Dietary intake, body weight, fecal/urine output, incisional healing, and subjective lameness were monitored routinely, and postoperative analgesia was provided for the first three days as needed (buprenorphine 0.02 mg/kg subcutaneously every twelve hours). At thirty days postoperatively, all animals were humanely euthanized via anesthetic overdose (isoflurane) and both Achilles tendons were harvested. The gross appearances of the left Achilles tendons were documented and photographed.

### 2.10. Histological Analysis

The tendon tissue was placed in a 10% formalin solution for 48 h, embedded in paraffin, cut into coronal sections, and stained. Each tendon sample underwent hematoxylin and eosin (H&E), Masson’s Trichrome (MT), and Gomori’s Reticulin (GR) staining. All the H&E and MT samples were imaged, randomized by rat, and graded as previously described (Figure 1) [7,8,57,58,59]. A minimum of six images (two at the tendon origin, two at the tendon mid-body, and two at the tendon insertion site) were acquired per stain per rat. Briefly, a histologic grading scale was employed that assessed the H&E sections on the criteria of fiber structure and arrangement, cellular density, cellular appearance, inflammation, neovascularization, and fatty deposits. The images were randomized and histological grading was conducted by one reviewer blinded to the treatment group assignment. Histological grades in each category were averaged for each rat and included in the statistical analysis as separate variables. The same grading methodology was repeated for the MT sections. The GR samples were randomized by rat and graded on the same scale, but only the variable of the fiber structure and arrangement was assessed.

### 2.11. Immunohistochemistry

The distribution and quality of the Collagen I deposition was assessed by immunohistochemical (IHC) staining. Briefly, coronal sections of tendon tissue were deparaffinized, rehydrated, and subjected to antigen retrieval using 0.25% trypsin at 37 °C for 30 min. After blocking with a 5% bovine serum, the sections were incubated with diluted anti-collagen type 1 (1:50; Bio Rad, Hercules, CA, USA) overnight. Goat anti-rabbit antibody served as the secondary antibody and all sections were counterstained with Nova Red, hematoxylin, and a bluing agent prior to dehydration and mounting. The sections were examined using bright field microscopy and imaged (Leica Microsystems, Wetzlar, Germany). The sections were quantitatively analyzed with the TWOMBLI macro for Fiji (ImageJ, Bethesda, MD, USA) [60,61].

### 2.12. Statistical Analysis

Immunofluorescence quantitative data was summarized for mean, standard deviation, median, and range, and the effects of growth factor exposure over time were evaluated using a two-way ANOVA. Categorical histological grades were summarized as counts and percentages. The effects of treatment on the histological grades were screened using a logistic regression analysis for a completely randomized design with subsamples. Numeric outcomes including the nuclear counts as a measure of tissue cellularity and TWOMBLI output metrics were summarized for the mean, standard deviation, median and range. The effects of treatment on the tissue cellularity and TWOMBLI output metrics were analyzed using a general linear model for a completely randomized design with subsamples. Ranked transformation was applied when the diagnostic analysis on the residuals exhibited a violation of normality and equal variance assumptions using Shapiro–Wilk test and Levene’s test. Post hoc multiple comparisons were performed with a Tukey’s adjustment. The statistical significance was identified at the level of 0.05. Analyses were conducted in SAS 9.4 TS1M7 for Windows 64x (SAS institute Inc., Cary, NC, USA).

## 3. Results

### 3.1. Rat Mesenchymal Stem Cell Isolation, Characterization, and Tenogenic Differentiation

#### 3.1.1. Isolation, Expansion, and Immunophenotyping of rAdMSCs and rBMSCs

The lineages of rAdMSCs and rBMSCs were successfully isolated and characterized with flow cytometry (Appendix A). An immunophenotypic surface marker analysis revealed that the rAdMSCs expressed CD29, CD73, CD 44H, CD90, and CD106, and they did not express CD11b/c, CD45, and CD31. Isolated rBMSCs expressed CD29 and CD106 and did not express CD11b/c, CD45, and CD31; however, widened peaks and mixed positive and negative expressions were noted on the CD73, CD44H, and CD90 analysis, suggesting a heterogenous population of cells at the tested passage.

#### 3.1.2. Cell Proliferation

Cellular proliferation of the rBMSCs and rAdMSCs was assessed over a period of 7 days using MTS assays, and the cellular viability was qualitatively assessed using calcein AM cellular staining (Figure 2). In the presence of the standardized growth medium provided, both cell types were viable and the cell numbers increased linearly over time. The proliferation rate and overall cellular viability of the rAdMSCs was superior to that of the rBMSCs so all further analyses and experimentation was conducted using rAdMSCs.

#### 3.1.3. Tenogenic Differentiation—Morphological Changes

Prior to exposure with enriched media, all wells were confirmed to contain adherent rAdMSC with minimal debris or cellular exfoliation. Twelve hours following exposure to the enriched media, marked changes in the cellular distribution and morphology were evident in all treatment groups (Figure 3). Cells exposed to TGF-β3 alone displayed apparent migration and clumping while maintaining adherence to the wells, while the cells exposed to CTGF/ascorbic acid (AA) and TGF-β3/CTGF/AA exhibited elongated, spindle-like morphology and, subjectively, a burst of proliferation. Two days following the initial exposure to enriched media, the TGF-β3/CTGF/AA cells exhibited distinct spindle morphology and had migrated into clumps with a mix of parallel and intersecting cell-to-cell alignment within the well; similar changes in the cell morphology and alignment were noted in the remaining two treatment groups at Day 3 of enriched media exposure (Figure 3). No apparent change in the TGF-β3/CTGF/AA cells was noted between Days 4 and 6 with the exception of a gradual, cumulative exfoliation of cells present and an accumulation of a granular extracellular byproduct visualized in all wells; a threshold of roughly 30% of cells exfoliated was reached on Day 6 of enriched media exposure and further culture of the TGF-β3/CTGF/AA group was halted. The cells exposed to TGF-β3 exhibited distinct spindle-morphology by Day 4 of culture and had migrated into independently oriented clumps of parallel-arranged cells; between Days 4 and 8 of culture, these cells remained relatively static but extracellular-matrix deposition (in the form of tendrils of extracellular material) could be visualized at the cell-to-cell contact points (Figure 3). The cells exposed to CTGF/AA remained static between Days 4 and 8 of culture, but beginning on Day 6, a granular extracellular byproduct was visualized in all wells, increasing in its amount over the remainder of the differentiation trial. Both the TGF-β3 and CTGF/AA wells reached >30% cellular exfoliation on Day 8 of culture and further culture was discontinued.

#### 3.1.4. Immunofluorescence

Tenogenic differentiation was characterized by immunofluorescence staining for cytoskeletal proteins, collagen I and F-actin, and for tendon-associated extracellular matrix (ECM) glycoproteins, tenascin C and tenomodulin. Morphological changes observed through the phase contrast light microscopy were confirmed with F-actin and collagen I immunofluorescence (Figure 4). Briefly, the cells treated with TGF-β3, CTGF/AA, and TGF-β3/CTGF/AA displayed filipodia extension, overall cell elongation, and parallel cytoskeletal conformation at the cell junctions on Days 1 and 4 of culture. Concurrently, a qualitative increase in collagen I expression was seen in all three groups with distinct points of high intensity fluorescence indicative of collagen bundle formation most apparent on Day 4. On Day 8, the cells treated with TGF-β3 remained adhered with a robust cytoskeletal morphology, characterized by parallel F-actin orientation with a high degree of cell-to-cell contact; however, the cells treated with CTGF/AA and TGF-β3/CTGF/AA displayed qualitatively poor cytoskeletal morphology on Day 8 of culture, characterized by a retraction of filipodia, reduction in cytoskeletal size, and an apparent decrease in cell numbers; the number of DAPI-stained nuclei significantly decreased between Day 1 and Days 4 and 8 of culture in the TGF-β3/CTGF/AA group (Figure 4, *p* < 0.001 and *p* < 0.001, respectively).

No significant differences in the proportion of cells displaying tenascin C immunofluorescence to the total cell counts were observed between the treatment groups on Days 1 and 4 (Figure 5); however, both the TGF-β3 and CTGF/AA groups exhibited increased tenascin C expression on Day 8 when compared to unexposed controls (*p* = 0.0185 and *p* = 0.0188, respectively). This can be attributed to a marked decrease in Tenascin C expression in the control cells compared to sustained upregulation in the TGF-β3 and CTGF/AA groups. In addition, cells exposed to TGF-β3 exhibited a unique but consistent tenascin C immunofluorescent signature characterized by high intensity fluorescence encircling the nucleus and moderate intensity fluorescence adjacent to the cell membrane; this signature was most apparent at Day 4 of culture but could be noted at all timepoints in the TGF-β3 group.

Rat AdMSC displayed a temporal upregulation of tenomodulin expression in response growth factor exposure (Figure 6). Cells exposed to CTGF/AA exhibited a significantly increased tenomodulin expression on Day 1 compared to the unexposed controls (*p* = 0.0420), and the relative magnitude expression remained consistent throughout the culture period. The cells exposed to TGF-β3 displayed a delayed but marked upregulation of tenomodulin expression on Day 4 (*p* < 0.001). Cells exposed to TGF-β3, CTGF/AA, and TGF-β3/CTGF/AA displayed a significantly greater tenomodulin expression compared to the unexposed controls on Day 8 (*p* = 0.0130, *p* = 0.0013, and *p* = 0.0041, respectively).

In summary, the in vitro data exhibiting changes in cell morphology, and an expression of tenogenic-specific protein markers, confirms that rat MSCs have tenogenic potential, and thus, can be induced into tenocyte-like cells under optimal chemical induction.

### 3.2. Intralesional Rat Mesenchymal Stem Cell Use in Achilles Tendon Injury

#### 3.2.1. Gross Morphology

On a gross morphologic analysis, the control tendons exhibited marked adhesions to surrounding soft tissue and overlying skin, and thick scar tissue overlying and fusing of the remaining medial and lateral thirds of the Achilles tendon was observed in four of six rats. Adhesions were rarely observed in the high dose and low dose groups (*n* = 1 and *n* = 0, respectively) and the overall appearance of the stem cell-treated Achilles tendons more closely resembled the uninjured controls with subjectively less scar tissue formation (see Figure 7). No evidence of infection or further tendon injury/rupture was noted in any operated limbs.

#### 3.2.2. Histological Analysis

At thirty days post-injury, all treatment groups displayed a marked hypercellularity, fiber disruption, and disorganization of the fiber arrangement compared to an uninjured tendon. Histological grading of both the H&E stained and Masson’s Trichrome (MT) stained tendon samples revealed no statistically significant differences in terms of the fiber structure and arrangement, cell density, cell appearance, inflammation, fatty deposits, or neovascularization grades (Figure 8); however, a qualitative review of the fiber arrangement and macroscopic structure revealed a marked matrix disruption with a complete loss of parallel structure and polar orientation of the matrix fibers in Achilles tendons treated with the fibrin gel alone (Figure 9). In addition, on a macroscopic scale, the Achilles tendons treated with both high and low doses of stem cells displayed more widespread neovascularization without concurrent perivascular inflammation or fat deposition.

Despite morphologic signs of healing, the Achilles tendons treated with high dose rAdMSCs, low dose rAdMSCs, and fibrin gel alone had significantly greater cellularity (mean cell counts) than the uninjured tendon (*p* ≤ 0.001 for each pairwise comparison). While the normal tendon samples were not included in the blinded histological analysis, macroscopic differences from the normal tendon in terms of cellularity, fiber pattern, and cell appearance were appreciated on a qualitative review (Figure 9).

#### 3.2.3. Immunohistochemistry

Results of the immunohistochemical staining for collagen I are presented in Figure 10. The TWOMBLI metrics of a high-density matrix (HDM, the proportion of pixels in an image corresponding to a stained matrix) and alignment (the extent to which fibers within the field of view are oriented in a similar direction) were considered for a quantification of the global collagen fiber pattern. No statistically significant differences in the quantitative analysis of Collagen I IHC were noted between the treatment groups.

## 4. Discussion

This study sought to evaluate the potential of ex vivo expanded and cryobanked allogenic rat MSCs for tenogenic differentiation in vitro and intralesional application in vivo. First, rat adipose-derived and bone marrow-derived stem cells were isolated, expanded, and characterized using flow cytometry. According to the International Society for Cellular Therapy, the minimum criteria for mesenchymal stem cells include cells that are plastic adherent in standard culture conditions, express CD106, CD73 and CD90, and must not express CD45, CD11b, or CD31 [14,62]. Additional MSC markers include CD29, which should be present in both bone marrow and adipose tissue, and CD44H, which is more specific to bone marrow but can be found in adipose-derived stem cells [62]. In the current project, the rAdMSC exhibited a >80% positive expression of CD 29, CD 73, CD 44H, CD 90, and CD 106 and a negative expression of CD 11 b/c, CD 45, and CD 31 (Appendix A), but the rBMSC only exhibited an apparent positive CD 29 and CD106 expression; while these differences could be a function of the cell source or the specific isolation procedure, the rBMSC flow cytometric results for CD 73, CD 44H, and CD 90 suggested a heterogenous cell line, as indicated by the bimodal positive and negative peaks for each of these cell surface markers [62]. As a result, <80% of the BMSCs expressed the stem cell markers, and hence, did not meet the criteria described by ISCT [14,62]. The rBMSC lineage also produced inferior results when compared to rAdMSC in a proliferation assay (Figure 2). Adipose-derived mesenchymal stem cells have been extensively utilized in tissue engineering due to their relative ease of acquisition and expansion, and positive results in terms of the tenogenic differentiation potential have been reported in rat, equine, and human cell lineages [19,25,26,62]; however, in a tenogenic differentiation trial testing the cell response to BMP-12-enriched differentiation media, similar isolation, characterization, and proliferation assays of three rat mesenchymal cell lines (rBMSC, rAdMSC, and rat synovial membrane MSCs) were conducted [43]. While all three lineages displayed an upregulation of tendon-associated markers including Scleraxis expression and Tnmd, TenC, and Col I production, the rBMSC displayed the highest degree of upregulation in response to BMP-12, and the authors concluded that rBMSC exhibited a superior tenogenic differentiation capability as compared to rAdMSC and rat synovial membrane MSCs [43]. In the current experiment, based on the suspected heterogenous nature of isolated rBMSC and apparent differences in the cellular proliferation, the decision was made to move forward with isolated rAdMSC for further use in vitro and in vivo.

Two potential tenogenic factors, TGFβ3 and CTGF, were chosen to evaluate their isolated and combined effects on rAdMSCs in a monolayer cell culture. The TGFβ family of cytokines plays key roles in cell proliferation and tissue morphogenesis during embryogenesis and a loss of TGFβ3 specifically (null mutants) directly correlates with a loss of normal tendon phenotype [20]. Several studies have reported on the tenogenic potential of TGFβ3 with a significant upregulation of tendon-associated markers including a Scleraxis and Mohawk expression and Tnmd, TenC, cartilage oligomeric matrix protein (COMP), and thrombospondin production in both two-dimensional and three-dimensional cultures [20,24,26,27,50]. One study compared the tenogenic potential of TGFβ1, TGFβ2, and TGFβ3 exposure on equine embryonic stem cells as compared to equine tenocytes; while all three TGFβ ligands upregulated the Scleraxis expression in vitro, further upregulation of tendon-associated glycoprotein and Col I production was only observed in embryonic stem cells cultured in TGFβ3 differentiation media [24]. Following an evaluation of isolated and combined exposure of rBMSC to TGFβ1, BMP-12, and CTGF/AA, Yin et al. proposed a stepwise tenogenic differentiation protocol that consisted of an isolated TGFβ1 stimulation for three days followed by a combined TGFβ1/CTGF/AA stimulation for an additional seven days [23]. The current study aimed to expand upon these previous experiments by testing the synergistic potential of TGFβ3 and CTGF/AA in a two-dimensional culture. Similar to the results described in Barsby et al., TGFβ3 was the most effective single tenogenic factor in terms of cell morphological changes, cytoskeletal structure/integrity, and tendon-associated glycoprotein production [24]. While morphological changes and an upregulation of TenC and Tnmd were detected in all groups within eight days of induction, a subjective loss of cell-to-cell connections (F-actin immunofluorescence) and cytoskeletal integrity (F-actin and Col I immunofluorescence) was observed between Days 4 and 8 of CTGF/AA and TGFβ3/CTGF/AA exposure (Figure 4).

Previous studies have identified several key cellular morphology and orientation changes that are associated with a tenogenic lineage including elongation of the cell into a fibroblast-like morphology, orientation along a polar axis, and a parallel alignment of clustered cells in a monolayer culture [17,19,25]. Several culture conditions have been described to directly enhance the MSC orientation and alignment during tenogenic differentiation, including dynamic mechanical stimulation such as cyclic uniaxial stretching, or culture environment manipulation such as a cellular attachment on an ECM scaffold or microgrooved matrix [19,63,64,65]. In a study assessing the effects of both a biochemical (TGFβ3 and BMP-12) and environmental (three-dimensional culture on decellularized tendon ECM scaffolds) stimulation of equine AdMSC toward a tenogenic lineage, TGFβ3 exposure in monolayer culture conditions elicited similar effects to the current study, including cellular elongation, migration resulting in parallel alignment, and marked proliferation (see Figure 3) [25]; concurrently, equine AdMSC cultured in TGFβ3-enriched media displayed a widespread upregulation of Col IA2, TenC, and Scleraxis expression [25]. However, the cellular alignment and uniform orientation along a polar axis was superior in scaffold cultures compared to their monolayer counterparts, and TGFβ3 exposure in a scaffold culture resulted in only a transient upregulation of Scleraxis and a concurrent upregulation of osteopontin, an osteogenic marker. In the current study, rAdMSC exposed to TGFβ3 in a monolayer culture exhibited a distinct spindle morphology by Day 4 of enriched media culture and migrated into independently oriented clumps of parallel aligned cells. It is probable that environmental stimuli, in the form of mechanical stimulation or a manipulation of the culture matrix, would enhance the observed tenogenic effects of TGFβ3 and yield a more uniform orientation and superior ECM organization in future tenogenic differentiation trials.

In contrast, cells in the CTGF/AA and TGFβ3/CTGF/AA, while notably exhibiting a spindle-like morphology earlier than cells in the TGFβ3 group, did not achieve the same degree of cellular alignment and organization (Figure 3). F-actin and Col I immunofluorescence was utilized to assess the cytoskeletal integrity, ECM organization, and overall cellular health of MSCs in this differentiation trial. The actin cytoskeleton plays a role in cell migration, cytokinesis, endocytosis, and the polarization and visualization of the interplay between filamentous actin networks in a cell can give insight into active cellular mechanisms such as division, migration, growth, or apoptosis [66]. Collagen Type I, while the primary component of the tendon matrix, is a non-specific, structural ECM component, and visualization of the Col I immunofluorescence will allow for a qualitative comparison of ECM deposition between treatment groups [16]. Cells treated with TGFβ3, CTGF/AA, and TGFβ3/CTGF/AA displayed filipodia extension, overall cell elongation, and parallel cytoskeletal conformation at the cell junctions on Days 1 and 4 of culture, but by Day 8 of culture, the cells exposed to CTGF/AA and TGFβ3/CTGF/AA exhibited a reduction in cytoskeletal size and a loss of apparent cell-to-cell connections. A qualitative review of the Col I immunofluorescence revealed superior collagen deposition in the TGFβ3 group with a robust cytoskeletal network persisting through Day 8 of culture. Most apparent as the green hyperintense fluorescent artifact in Figure 5 and Figure 6 (CTGF/AA and TGFβ3/CTGF/AA groups, respectively), a granular extracellular byproduct was observed in all wells exposed to CTGF/AA, either alone or in combination. This result was not described in previous studies utilizing CTGF/AA in tenogenic differentiation, and further characterization of the byproduct was outside of the scope of this project [15,23]; however, detrimental effects of byproduct accumulation cannot be ruled out as a contributing factor to the poor survival and relatively inferior tenogenic differentiation results in this group.

Immunofluorescent staining and quantification of the glycoproteins, tenomodulin and tenascin C, served as the downstream markers of effective tenogenic differentiation. Tenascin C functions in tendon ECM to aid collagen fiber alignment and orientation [17]; other tenogenic differentiation trials have reported a transient burst of TenC production in the first few days of differentiation, followed by a gradual decline in the ECM concentration for the remainder of culture [50,67]. Tenomodulin is a transmembrane glycoprotein vital to tenocyte proliferation and tendon maturation [17]; it is a downstream product of the pathway activated by the Scleraxis transcription factor, a well-accepted marker of tenogenic lineage, and increasing the tenomodulin concentrations in vitro are directly correlated to Scleraxis upregulation [68,69]. In the current experiment, a significant upregulation of TenC immunofluorescence was noted on Day of 8 culture in the TGFβ3 and CTGF/AA groups compared to the matched controls, and a significant upregulation of the Tnmd immunofluorescence was first noted on Days 1, 4, and 8 in the CTGF/AA, TGFβ3, and TGFβ3/CTGF/AA groups, respectively. In addition, cells in the TGFβ3 group exhibited a unique but consistent tenascin C immunofluorescent signature characterized by a high intensity fluorescence encircling the nucleus and moderate intensity fluorescence adjacent to the cell membrane, first noted on Day 4 of culture. Similar patterns and relative intensity of TenC immunofluorescence have been documented in other tenogenic differentiation or tenocyte characterization trials [70,71], and in an experiment assessing the effect of a BMSC coculture with mechanically stretched ligament fibroblasts, a hyperintense TenC immunofluorescent signature was directly correlated with TenC mRNA expression, quantified using a reverse-transcription-quantitative polymerase chain reaction (RT-qPCR) [72]. Therefore, based on superior tendon-associated glycoprotein expression upregulation and appropriate morphologic, behavioral, and proliferative changes, we can conclude that TGFβ3 alone was the most effective tenogenic factor in this experiment, and we can confirm that the rAdMSC lineage can be effectively directed toward a tenogenic lineage in vitro, thus, confirming its tenogenic potential.

Following characterization of the rAdMSC and confirmation of their tenogenic potential in experimental conditions, we sought to evaluate the rAdMSC therapeutic potential in vivo in an acute tendon injury model. Fibrin gel (fibrinogen/factor XIII combined with thrombin/calcium chloride) was selected as the delivery vehicle for intralesional MSC therapy to ensure the safety and proper localization of delivered cells within the injured tissue [11]. Widely used in cardiac, neurological, and plastic surgery, fibrin gel is a biocompatible, biopolymeric substrate that allows for cell suspension for delivery and facilitates cell adhesion, proliferation, and differentiation after implementation [50,53,73]. To allow for accurate evaluation of the effect of the rAdMSC dose in the current experiment, the control rats underwent Achilles injury as described and received fibrin gel alone without MSCs; a potential therapeutic benefit of fibrin gel cannot be ruled out and serves as a limitation to the current study.

Thirty days following a left Achilles injury and intralesional rAdMSC implantation, all rats were euthanized, and the left Achilles tendons were evaluated for gross pathology, structurally assessed using histopathology, and compositionally assessed using immunohistochemistry. On the gross morphologic analysis, the control tendons exhibited marked adhesions to the surrounding soft tissue and overlying skin, and an apparent thickening of the remaining Achilles tendon was noted in four of six control rats; adhesions were rarely observed in rAdMSC-treated tendons and the overall appearance of the stem cell-treated Achilles tendons more closely resembled the uninjured controls with subjectively less scar tissue formation. Tendon has a comparatively poor capacity for repair following injury, and primary healing is often prolonged and scar mediated, as opposed to regenerative [16]. Manifestations of scar tissue, adhesions, or fibrotic scars on gross pathology give insight into ineffective remodeling on a microscopic scale, including a lack of compartmental collagen organization, chronic tissue inflammation, and the loss of gliding functionality [74]. Qualitative structural and compositional differences were noted on the histopathology, including a marked matrix disruption with a complete loss of parallel structure and polar orientation of the matrix fibers in the control tendons, and notable neovascularization without a concurrent perivascular inflammation or fat deposition in the rAdMSC-treated tendons. However, no significant differences in the histological grades or quantitative assessment of the Col I deposition (IHC) were noted between the rAdMSC-treated and control tendons, regardless of the rAdMSC dose. Several reports of intralesional MSC therapy in acute tendon injury models corroborate these findings [6,7,54,75,76]; for example, in an analysis of the therapeutic potential of stem-cell-derived extracellular vesicles as compared to rBMSCs in a rat Achilles defect model, tendons treated with rBMSCs alone yielded no significant difference in the histological grade or quantitative Col I:Col III ratio from the control tendons [8]. In a chronic model of equine superficial digital flexor tendon injury, an intralesional injection of autologous AdMSC suspended in serum (three weeks after lesion induction) yielded no significant differences from the controls in terms of gross histology, histological grade, vascularization, or Col I deposition at 12- and 24-weeks post-injury [77]. Similar to previous publications, no effect of MSC dose was appreciated in this experiment, either on the gross morphology or histopathologic evaluation [28,29,31]; this challenges the notion that more cells are indeed better, and in this experiment, the suspension and delivery of 4 × 10^6^ rAdMSC into a subjectively small lesion without a significant cellular loss or compromise was difficult. Saether et al. noted a MSC-dose dependent difference in the innate immune response characterized by higher concentrations of pro-inflammatory cytokines and M1 macrophages in rat medial collateral ligaments treated with high dose (4 × 10^6^) MSCs as compared to those treated with low dose (1 × 10^6^) MSCs [28]. It has been suggested that the immunomodulatory potential of MSCs may not be unconditional, and that high doses of MSCs can still induce an immune response through MCH II upregulation in certain environmental conditions or direct stimulation through a toll-like receptor recognition of pathogen-associated molecular patterns [13,14]. Based on this and previous studies, we suggest that increasing the MSC dose may not be therapeutically beneficial.

In summary, we report that despite the potential to undergo tenocyte differentiation in vitro, the rAdMSCs isolated and characterized in this study did not demonstrate a significant increase in tendon healing in vivo. The in vivo study did confirm the potential of fibrin glue as a delivery scaffold for MSCs, but the data suggests that an additive therapy which might include adding specific growth factors or using a delivery device distinct from fibrin glue may be needed to trigger tenocyte differentiation for the rAdMSCs used. Future experiments to design such strategies may be needed.

## 5. Conclusions

Rat adipose-derived and bone marrow-derived stem cell lineages were isolated and compared using flow cytometry for stem cell surface markers and using proliferation (MTS) assays. Rat AdMSC displayed a superior proliferation and more homogenous CD 73, CD 44H, and CD 90 expression as compared to rBMSC. The tenogenic differentiation potential of the rAdMSC lineage was tested through isolated and combined TGFβ3 and CTGF/AA stimulation, and the most effective tenogenic factor in terms of the cellular morphology change, cell alignment/orientation, sustained cellular viability, and tendon-associated glycoprotein upregulation was TGFβ3. The therapeutic potential of undifferentiated rAdMSC was assessed using a rat acute Achilles tendon injury model; on gross morphology, rAdMSC-treated tendons exhibited fewer adhesions and subjectively less scar tissue and tissue thickening, but no significant differences in the histological grade or tissue Col I deposition was noted between the rAdMSC and control tendons. No effect of the mesenchymal stem cell dose was noted on the gross pathology or histopathological analysis.

## Figures and Tables

**Figure 1 bioengineering-09-00407-f001:**
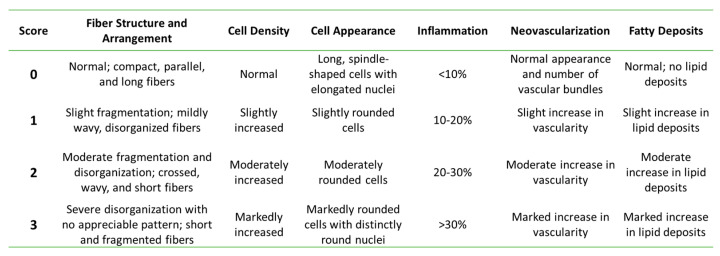
Histological grading scale as described by de Girolamo et al. (2019) [59].

**Figure 2 bioengineering-09-00407-f002:**
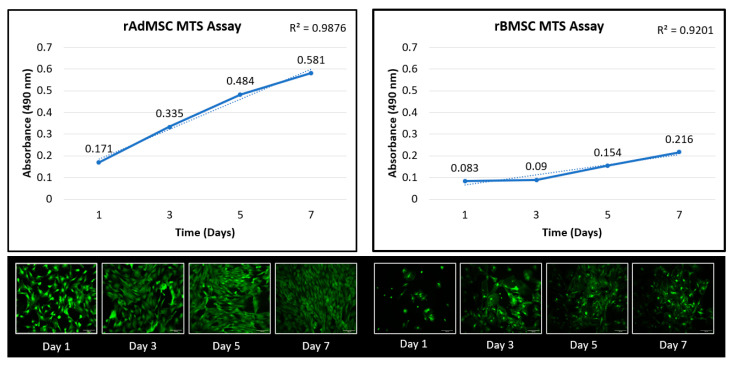
Proliferation and viability assays of rat adipose-derived and bone-marrow-derived mesenchymal stem cells. The solid line represents average absorbance at 490 nm at Days 1, 3, 5, and 7, and the dashed line represents the linear trendline for the respective data; the coefficients of determination (R^2^ values) of linear regression are reported in the upper right corner of each graph. Representative confocal images of calcein AM stained cells at each timepoint are included below the respective cell lineage’s graph. Scale bar = 100 µm.

**Figure 3 bioengineering-09-00407-f003:**
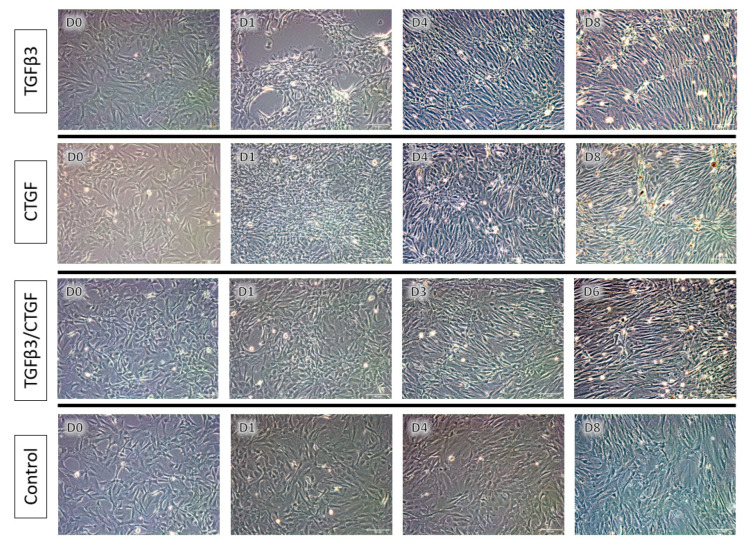
Tenogenic differentiation trial, bright light phase-contrast microscopy. rAdMSCs were maintained in complete medium alone (Control) or were exposed to either TGF-β3 (10 ng/mL), CTGF (100 ng/mL) with 50 µg/mL ascorbic acid, or TGF-β3 (10 ng/mL), CTGF (100 ng/mL), and 50 µg/mL ascorbic acid. Representative images from Days 0, 1, 4, and 8 of the Control, TGF-β3, and CTGF groups and Days 0, 1, 3, and 6 of the TGF-β3/CTGF group are depicted above to illustrate the change in cellular morphology, orientation, alignment, and extracellular matrix. Scale bar = 100 µm.

**Figure 4 bioengineering-09-00407-f004:**
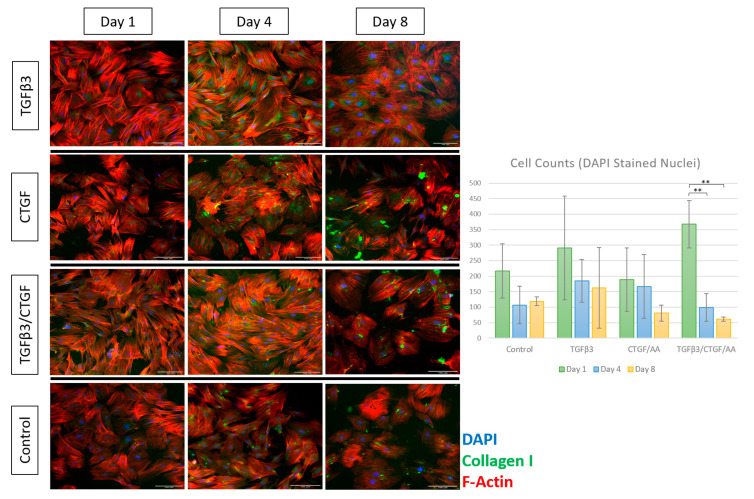
Expression of collagen Type I (green) and F-actin (red) in rAdMSCs exposed to TGF-β3 (10 ng/mL), CTGF (100 ng/mL) with 50 µg/mL ascorbic acid, or TGF-β3 (10 ng/mL), CTGF (100 ng/mL), and 50 µg/mL ascorbic acid at Days 1, 4, and 8 of culture; cell nuclei counterstained with DAPI (blue). Mean values of DAPI-stained nuclei counts are depicted in the accompanying chart; cell counts significantly decreased between Day 1 and Days 4 and 8 in the group treated with TGFβ3/CTGF/AA. Scale bar = 100 µm. “**” represents *p* < 0.01 in pairwise comparison. Figure 1. Histological grading scale as described by de Girolamo et al. (2019) [59].

**Figure 5 bioengineering-09-00407-f005:**
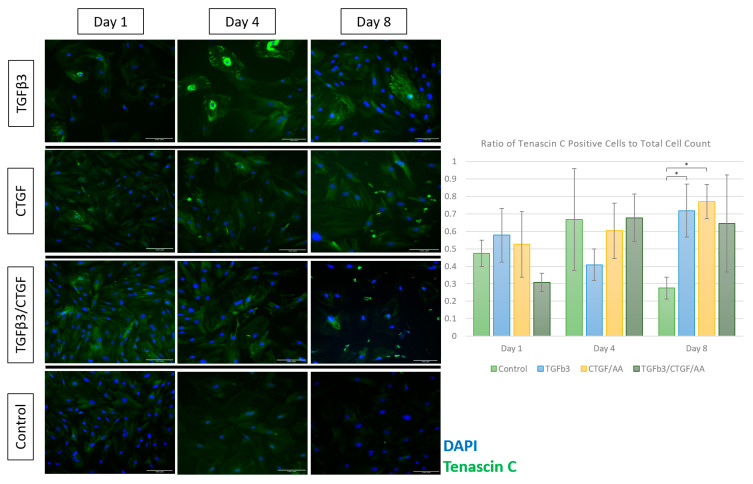
Expression of tenascin C (green) in rAdMSCs exposed to TGF-β3 (10 ng/mL), CTGF (100 ng/mL) with 50 µg/mL ascorbic acid, or TGF-β3 (10 ng/mL), CTGF (100 ng/mL), and 50 µg/mL ascorbic acid at Days 1, 4, and 8 of culture; cell nuclei counterstained with DAPI (blue). Immunofluorescence staining of tenascin C was significantly upregulated in the groups treated with CTGF/ascorbic acid (AA) and with TGFβ3 on Day 8. Scale bar = 100 µm. “*” represents *p* < 0.05 in pairwise comparison.

**Figure 6 bioengineering-09-00407-f006:**
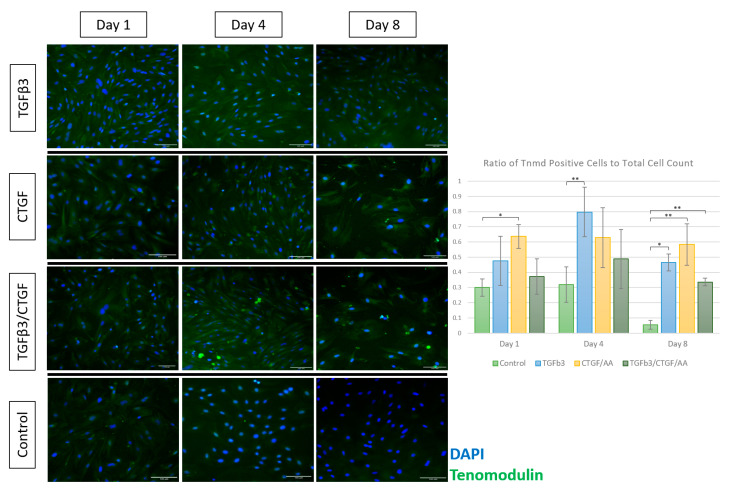
Expression of tenomodulin (green) in rAdMSCs exposed to TGF-β3 (10 ng/mL), CTGF (100 ng/mL) with 50 µg/mL ascorbic acid, or TGF-β3 (10 ng/mL), CTGF (100 ng/mL), and 50 µg/mL ascorbic acid at Days 1, 4, and 8 of culture; cell nuclei counterstained with DAPI (blue). Immunofluorescence staining of tenomodulin was significantly upregulated in the group treated with CTGF/ascorbic acid (AA) on Day 1, in the group treated with TGFβ3 on Day 4, and in the groups treated with CTGF/AA, TGFβ3, and TGFβ3/CTGF/AA on Day 8. Scale bar = 100 µm. “*” represents *p* < 0.05 and “**” represents *p* < 0.01 in pairwise comparison.

**Figure 7 bioengineering-09-00407-f007:**
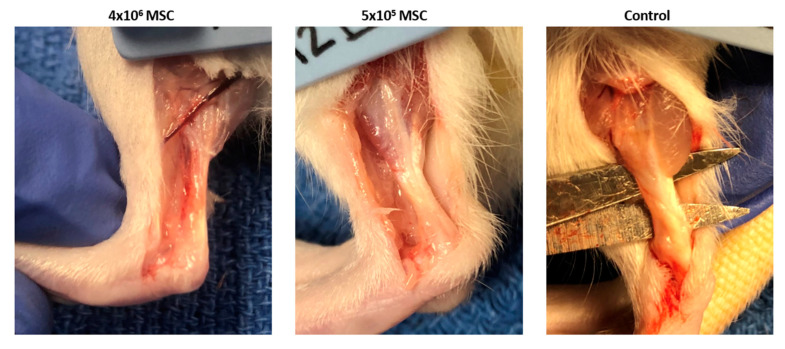
Gross appearance of the rat Achilles tendon 30 days after tendon injury and intralesional rAdMSC treatments at the indicated dosages. Tendons were assessed for size, presence of adhesions/scar tissue, relative vascularity, lesion visibility, and subjective assessment of repair.

**Figure 8 bioengineering-09-00407-f008:**
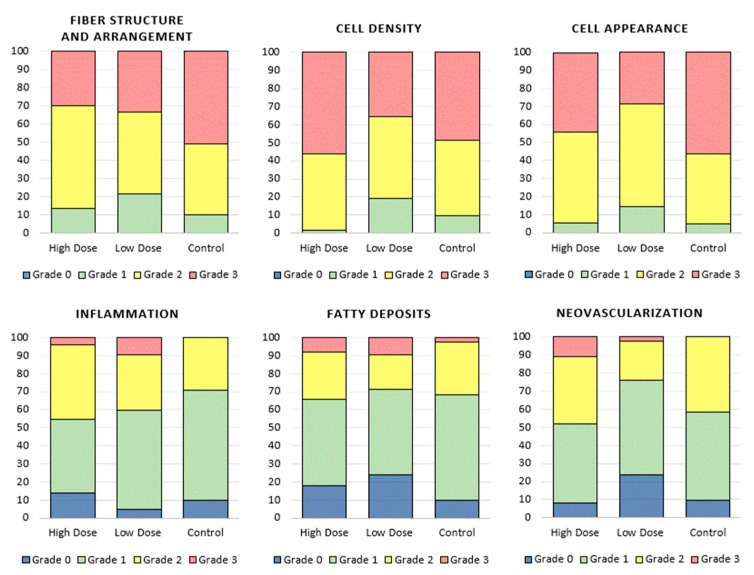
Results of histological grading (H&E) in terms of fiber structure, cell density, cell appearance, inflammation, neovascularization, and fatty deposits; color corresponds to histological grade and color bar height indicates the percentage of the indicated treatment group assigned that histological grade. No statistically significant differences in histological grade were detected between the high dose, low dose, and control groups in any category.

**Figure 9 bioengineering-09-00407-f009:**
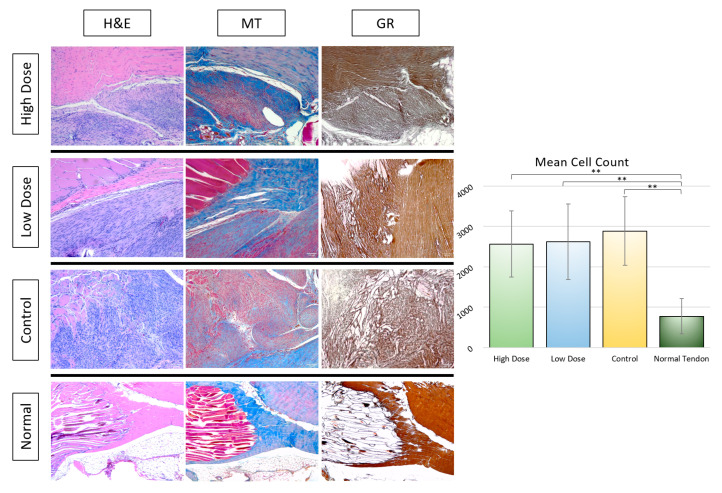
Histological staining of rat Achilles tendon 30 days after tendon injury and treatment compared to uninjured rat Achilles tendon. H&E staining highlights cellular density, architectural changes to the tendon matrix, cellular morphology, and abnormal vascularization or fat deposition within the tissue. Masson’s Trichrome staining highlights changes in the tendon extracellular matrix with more normal tendon tissue (blue) and muscle (bright red) differentiated from the abnormal matrix (dull red). Gomori’s Reticulin stain highlights abnormal reticular fiber conformation and overall organization of fibrous tissue within the tendon. Scale bar = 100 µm. Mean cell counts on H&E-stained samples are depicted to the right. “**” represents *p* < 0.01 in pairwise comparison.

**Figure 10 bioengineering-09-00407-f010:**
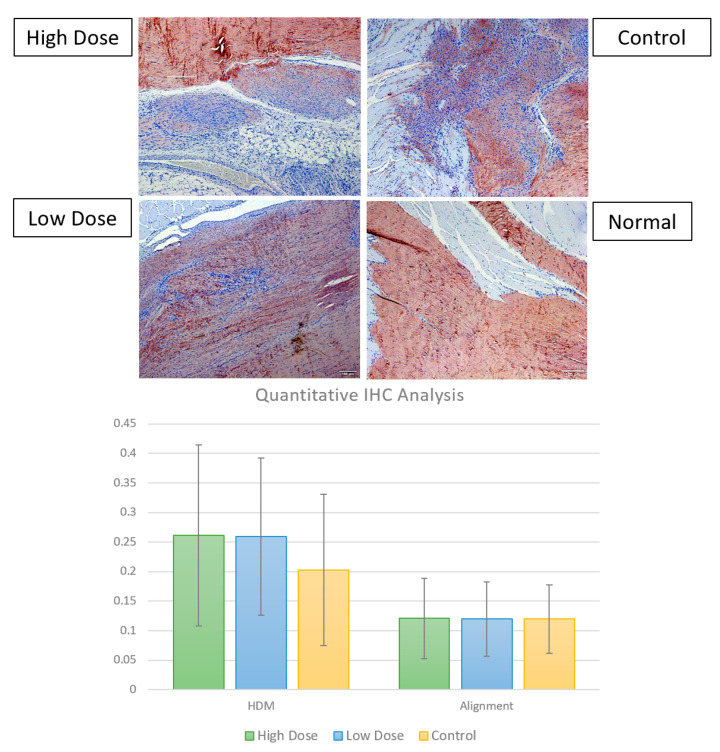
Collagen Type I immunostaining of rat Achilles tendon 30 days after tendon injury and treatment compared to normal, uninjured rat Achilles tendon. Scale bar = 100 µm. Quantitative TWOMBLI metrics of the high-density matrix and matrix alignment for the high dose, low dose, and injured control groups are presented in the chart above, conveyed as ratios of high-density matrix to total matrix and ratios of aligned fibers to total fiber content, respectively. No significant differences in the quantitative analysis of collagen I IHC were noted between treatment groups.

## Data Availability

Not Applicable.

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
