# Peer review of "Mesenchymal Stem Cell Use in Acute Tendon Injury: In Vitro Tenogenic Potential vs. In Vivo Dose Response"

_bioengineering, 2022, doi:10.3390/bioengineering9080407_

Round 1
Reviewer 1 Report
Changing the title is already a step toward better understanding by the readers. What for me is not consequent is that your paper shows 2 different aspects that are not bound in the experimental approach.
I think this paper can be published as it is but is of low interest for scientific community working in cell therapies.
Reviewer 2 Report
I appreciate the efforts made by the authors to revise the manuscript.
The title is, now, representative of the article content.
Lines 102-107 give a good explanation of how the in vitro relates to the in vivo, having in mind those limitations.
Minor changes:
- Add the number of replicates in the legend of each graph.
This manuscript is a resubmission of an earlier submission. The following is a list of the peer review reports and author responses from that submission.
Round 1
Reviewer 1 Report
Allogenic rat adipose-derived MSCs were subjected to tendon differentiation and their potential to treat acute tendon injury in rats was evaluated.
The characterization of BMSCs was not indicative of a homogenous stem cell population and the authors went forward with AdMSCs.
Major changes:
-The significant increase in Tenascin C on day 8 in the treated groups is mainly due to the decrease in Tenascin C in the control group rather than a difference compared to day 1. This is more evident for the Tenomodulin results where almost no difference between day 1 and day 8 exhist. I would suggest complementing the evaluation with the expression of those genes by RT-PCR and comparing the control of MSCs at day 0 with the treated groups on day 8.
Minor changes:
-The literature was well described in the manuscript however some citations/articles description could be waived especially in the discussion section.
-Results sections should have a paragraph, just a sentence, at the end resuming the important finding.
-In figure 4 the ascorbic acid should be labeled also in the image (CTGF/AA and TGFb3/CTGF/AA)
Figure 4 should also include the quantification of Collagen and F-Actin intensity.
- Figure 7 should indicate what was measured in the gross morphology of the rat tendon.
- It is strongly suggested to highlight the novelty of this paper.
Author Response
Comments and Suggestions for Authors
Allogenic rat adipose-derived MSCs were subjected to tendon differentiation and their potential to treat acute tendon injury in rats was evaluated.
The characterization of BMSCs was not indicative of a homogenous stem cell population and the authors went forward with AdMSCs.
Major changes:
-The significant increase in Tenascin C on day 8 in the treated groups is mainly due to the decrease in Tenascin C in the control group rather than a difference compared to day 1. This is more evident for the Tenomodulin results where almost no difference between day 1 and day 8 exist. I would suggest complementing the evaluation with the expression of those genes by RT-PCR and comparing the control of MSCs at day 0 with the treated groups on day 8.
We would like to thank the reviewer for this comment. We agree that complementing our protein expression data with genetic expression would be valuable, and we will incorporate that suggestion in our future research. For this experiment, we focused on protein expression (production) in both the in vitro and in vivo methodology to highlight the importance of extracellular matrix quality and composition in tendons; our overarching goal was to critically evaluate the use of intralesional mesenchymal stem cell implantation. The in vitro results will aid future research in the establishment of a tenogenic differentiation protocol, but we acknowledge that further research including complementary gene and protein expression quantification techniques will add value to future works.
Minor changes:
-The literature was well described in the manuscript however some citations/articles description could be waived especially in the discussion section.
Thank you for this suggestion. The citations have been re-evaluated, particularly in the discussion.
-Results sections should have a paragraph, just a sentence, at the end resuming the important finding.
We thank the reviewer for this suggestion. The results sections have been evaluated and edited for clarity.
-In figure 4 the ascorbic acid should be labeled also in the image (CTGF/AA and TGFb3/CTGF/AA)
Figure 4 should also include the quantification of Collagen and F-Actin intensity.
Thank you for catching this omission. The ascorbic acid label has been added.
Thank you for your suggestion regarding collagen and F-actin quantification; that particular set of immunofluorescent stains was elected to assess the stem cells’ cytoskeletal and cell-to-cell communication integrity among culture conditions. Quantification of F-Actin and collagen immunofluorescence posed a significant challenge as the cells reached confluence. Automated quantification was unable to consistently and accurately distinguish cell borders in the images and fluorescence quantification of the images yielded as variable results. Therefore, interpretation of the stains was limited to qualitative review of cytoskeletal conformation.
- Figure 7 should indicate what was measured in the gross morphology of the rat tendon.
Thank you, the description has been added.
- It is strongly suggested to highlight the novelty of this paper.
We thank the reviewer for this suggestion. This research is compelling because we detected no dose-dependent response of intralesional allogenic mesenchymal stem cells in acute tendon injury in vivo, despite promising extracellular matrix production in vitro. This suggests that there was no advantage to providing higher concentration of cells relative to the extent or quality of lesion healing.
Reviewer 2 Report
The subject is actual and intersting for the scientific and medical community but this paper fails on some aspects:
1. Differentiation into tenocytes is an open question on how this should be done and the results shown by the authors are not enough convincing on this differantiation.
2. Following point 1 is of course difficult then to see a clear effect in vivo.
3. I didn't undestand also why the authors used undufferentitaed AdMSC for the in vivo test
In general I would suggest to better differentiate cells if possible but this is the challenge today or alternatively to use adipose-derived stromal cascular fraction for the entire study.
Author Response
The subject is actual and interesting for the scientific and medical community but this paper fails on some aspects:
- Differentiation into tenocytes is an open question on how this should be done and the results shown by the authors are not enough convincing on this differentiation.
We would like to thank the reviewer for this comment. We agree further research regarding protocols for tenogenic differentiation are needed. Multimodal cellular stimulation beyond growth factor exposure (ie. mechanical stimulation, three-dimensional culture conditions, etc.) may yield more consistent results. For this experiment, we focused on extracellular protein production in both the in vitro and in vivo methodology to highlight the importance of extracellular matrix quality and composition in tendons. We hope that the in vitro results will inform future research in the establishment of a tenogenic differentiation protocols including complementary gene and protein expression quantification.
- Following point 1 is of course difficult then to see a clear effect in vivo.
Please see Response #3.
- I didn't undestand also why the authors used undufferentitaed AdMSC for the in vivo test.
In general I would suggest to better differentiate cells if possible but this is the challenge today or alternatively to use adipose-derived stromal cascular fraction for the entire study.
Thank you for your comments. We selected undifferentiated cells for use in this initial trial for the following reasons. First, we wished to assess the intralesional therapeutic potential of undifferentiated AdMSC. Also, we wished to evaluate the effect of allogenic stem cell dose on tendon healing. This research is compelling because we detected no dose-dependent response of intralesional allogenic mesenchymal stem cells in acute tendon injury in vivo, despite promising extracellular matrix production in vitro.
Round 2
Reviewer 1 Report
The authors made an effort to assess the reviewers suggestions. However the article has important issues that have not been tackled.
We are not observing tendon differentiation in vitro and the text description lead us to a mistake.
There is no link between the differentiation into tendon in vitro and the use of undifferentiated MSCs in vivo. There is a problem related to the experimental design and the long text and long descriptions (results and discussion sections) misdirected the attention from the simple main conclusions
In my opinion the article is assigning an excessive value to the results which are often overvalued.
Reviewer 2 Report
After considering the author's point of view I would suggest changing the title of the manuscript because if you read the title now it seems that the authors treat the Achilles tendon after differentiating the cells into tenocytes which is not true. This was my main concern for this paper. If they agree to change the title it could be considered for publication but, of course, the argument loses a lot of interest for the scientific community because these aspects have been already widely inspected by researchers.